# Clear Cell Sarcoma (CCS) of the Soft Tissue: An Update Narrative Review with Emphasis on the Utility of PRAME in Differential Diagnosis

**DOI:** 10.3390/jcm14041233

**Published:** 2025-02-13

**Authors:** Gerardo Cazzato, Francesco Piscazzi, Alessandra Filosa, Anna Colagrande, Paolo Del Fiore, Francesca Ambrogio, Chiara Battilotti, Andrea Danese, Serena Federico, Fortunato Cassalia

**Affiliations:** 1Section of Molecular Pathology, Department of Precision and Regenerative Medicine and Ionian Area (DiMePRe-J), University of Bari “Aldo Moro”, 70124 Bari, Italy; anna.colagrande@gmail.com; 2Dermatology Unit, IRCSS Humanitas Research Hospital, 20089 Rozzano, Italy; francesco.piscazzi@humanitas.it; 3Department of Biomedical Sciences, Humanitas University, 20072 Pieve Emanuele, Italy; 4Pathology Department, “A. Murri” Hospital-ASUR Marche, Aree Vaste n. 4 and 5, 63900 Fermo, Italy; alessandrafilosa@yahoo.it; 5Soft Tissue, Peritoneum and Melanoma Surgical Oncology Unit, Veneto Institute of Oncology IOV-IRCCS, 35128 Padua, Italy; paolo.delfiore@iov.veneto.it; 6Section of Dermatology and Venereology, Department of Precision and Regenerative Medicine and Ionian Area (DiMePRe-J), University of Bari “Aldo Moro”, 70124 Bari, Italy; dottambrogiofrancesca@gmail.com; 7Unit of Dermatology, Department of Clinical Internal, Anesthesiological and Cardiovascular Sciences, Sapienza University, 00185 Rome, Italy; chiara.battilotti@gmail.com; 8Section of Dermatology and Venerology, Department of Medicine, University of Verona, 37129 Verona, Italy; adanese4@gmail.com; 9Unit of Dermatology, Department of Health Sciences, Magna Grecia University, 88100 Catanzaro, Italy; drserenafederico@gmail.com; 10Dermatology Unit, Department of Medicine (DIMED), University of Padua, 35121 Padua, Italy; fortunato.cassalia@studenti.unipd.it

**Keywords:** clear cell sarcoma, CCS, differential diagnosis, melanoma, PRAME, diagnostics

## Abstract

Clear Cell Sarcoma (CCS) of soft tissue is a rare and highly malignant neoplasm primarily affecting young adults, often presenting in the deep soft tissues of the extremities. Despite morphological and immunophenotypic similarities to melanoma, CCS arises from connective tissues and is characterized by a distinct genetic hallmark: the EWSR1-ATF1 fusion resulting from t(12;22)(q13;q12) translocation. This genetic signature is absent in melanoma, making molecular diagnosis essential for accurate differentiation. Additionally, recent evidence highlights the utility of PRAME as an immunohistochemical marker to distinguish CCS from melanoma and other neoplasms. Clinically, CCS commonly involves tendons and aponeuroses, with metastatic potential leading to poor prognoses despite optimal local disease management. Histologically, CCS features lobular growth, spindle-to-epithelioid cells with clear cytoplasm, and low mitotic activity, often necessitating a multimodal diagnostic approach incorporating histopathology, immunohistochemistry, and molecular testing. Therapeutically, wide surgical excision remains the cornerstone for localized disease, with sentinel lymph node biopsy aiding in staging. Adjuvant radiotherapy is considered in select cases, while chemotherapy has limited efficacy in metastatic settings. Emerging treatments, including targeted therapies focusing on EWSR1-ATF1-driven pathways and immune checkpoint inhibitors, offer hope for improved outcomes. This review synthesizes current knowledge on CCS, emphasizing diagnostic challenges, the role of PRAME, and advancements in therapeutic strategies to enhance patient care.

## 1. Introduction

Clear Cell Sarcoma (CCS) of soft tissue is a rare, distinctive, and highly malignant neoplasm that predominantly affects young adults and typically presents within deep soft tissues of the extremities, with a clinical appearance of a palpable mass, sometimes with pain and tenderness [1]. Although the lesion is extremely large, usually the overlying skin and dermis are not involved and, exceptionally, genetically confirmed CCS may occur in the skin [1]. Its histopathological features and immunohistochemical profile closely resemble those of melanoma, often leading to significant diagnostic challenges [1,2]. CCS was first described by Enzinger in 1965 also as malignant melanoma of the soft tissues due to its already mentioned morphological and immunophenotypic overlap with melanoma [2], but, unlike melanoma which primarily arises from melanocytes within the skin, CCS arises in connective tissue structures, such as tendons and aponeuroses [3]. In the World Health Organization (WHO) *Classification of Soft Tissue and Bone Tumors*, Fifth Edition, CCS is recorded with code 9044/3 of the *International Classification of Diseases for Oncology* (ICD-O) and the designation as “malignant melanoma of the soft tissues” is not recommended [4]. In any case, both CCS and melanoma share a common origin from neural crest cells, explaining their shared expression of markers such as S100 protein and HMB-45 [4]. Despite these similarities, CCS possesses unique molecular features, such as t(12;22)(q13;q12) translocation, that fuses EWSR1 on chromosome 22 with ATF1 (member of the CREB transcription factor family on chromosome 12), a chromosomal rearrangement absent in melanoma and in other soft tissue sarcomas [5]. This genetic signature has become crucial in differentiating CCS from melanoma, considering that molecular diagnosis is often necessary to discriminate between these lesions. More recently, a new marker named PReferentially expressed Antigen in MElanoma (PRAME) was reported as a potentially useful diagnostic tool to discriminate dysplastic naevus from malignant melanoma, and, although it can be positive in other mesenchymal neoplasms such as synovial sarcoma, myxoid liposarcoma, and neuroblastoma [6], it is relatively rare in CCS, being able to provide pathologists with an immunohistochemical tool to reduce diagnostic ambiguity [7]. In more detail, PRAME was first isolated by autologous T-cells clone of a patient with metastatic cutaneous melanoma and in recent years was employed as marker to help dermatopathologists with the differential diagnosis between high-grade dysplastic nevus and melanoma in situ, and beyond that with other types of melanocytic lesions [7,8]. Furthermore, a recent study showed that PRAME was significantly up-regulated in a large number of tumors, and its high expression was correlated with poor overall survival across many cancer types, based on a pan-cancer survival analysis. Additionally, in a number of malignancies, PRAME expression levels were closely associated with immunological neoantigens, immune infiltration, immune score, immune checkpoint, tumor mutation burden, microsatellite instability, mismatch repair, and DNA methyltransferase. Finally, PRAME was linked to the control of many signaling pathways implicated in tumor immunity and tumorigenicity, according to GSEA analysis [9].

Taking into account the most up-to-date knowledge about the topic, this review will discuss the clinical presentation, histological characteristics, and molecular diagnostics of CCS, emphasizing the role of PRAME in distinguishing CCS from melanoma and other histologically similar neoplasms. Finally, it will perform an analysis of the most up-to-date therapeutical modalities useful to treat patients affected by CCS.

## 2. Material and Methods

Literature research was conducted on PubMed, Web of Science (WoS), MEDLINE, Google Scholar, and Cochrane, until November 2024, using keywords: “clear cell sarcoma”, “CCS”, and “malignant melanoma of soft tissue” together with “PRAME”. Four authors conducted the research independently, duplicates were eliminated, and articles were collegially discussed, with particular attention to the role of PRAME in differential diagnosis between CCS and melanoma. Furthermore, backward literature research was conducted on selected articles.

## 3. Results

### 3.1. Clinical Features: Presentation, Disease Course, and Prognosis

Clinically, CCS affects the upper limbs in about 25% of cases and tumors tend to grow around tendons, are usually less than 3 cm in diameter, and are occasionally painful. Females and males are equally affected (with a slight female predominance) with age between 20 and 40 years (median: 30 years) and approximately 40% of cases occur on the foot and ankle [8] with another 30% involving the knee, thigh, and hand [5,8,9,10]. Despite optimal management of local disease, a large proportion of patients develop recurrent or metastatic disease: about 50% of patients develop metastatic disease, often many years after the initial diagnosis. The 5- and 10-year survival rates are 52% and 25%, respectively [2], and survival rate is significantly lower for patients with stage 3 or 4 disease at diagnosis [3]. Tumor size appears to be associated with worse prognosis [4], but it is not clear whether purely dermal tumors have a better prognosis than those deeply located. Furthermore, a recent study addressed the issue of prognostic markers of CCS, analyzing data from the Surveillance, Epidemiology, and End Results (SEER) database, representing approximately 30% of the US population, over the time period 1973–2009. In particular, the authors extrapolated the demographic and clinical-pathological data of 175 patients diagnosed with CCS, evaluating a 5-year survival rate of 62.9% and 10-year survival rate of 51.3%. According to these data, 98 patients (56.0%) were men and 77 patients (44.0%) were women, with 101 patients (57.71%) with localized CCS, 59 patients (33.71%) with regional CCS, and 15 patients (8.57%) with distant/metastatic CCS. Most common location at diagnosis was lower extremities (74.29%) and the tumor size most commonly was ≤3 cm at the time of diagnosis (44.57%) with 174 patients (93.71%) undergoing surgical resection. Performing univariate Cox regression analysis, factors such as race, sex, tumor stage, tumor size, chemotherapy, and surgery were statistically significant, but with multivariate Cox regression analysis only size and tumor stage were significant, with a higher risk of death among patients with distant stage and regional stage and patients with tumor size >3 cm [10].

Table 1 summarizes the most important clinical and pathological features of CCS.

### 3.2. Diagnosis: Histopathology, Immunohistochemistry, and Molecular Genetics

Usually, the diagnosis is made taking into account histopathological and immunohistochemical features with molecular confirmation. Macroscopically, lesions are often necrotic, with cystic change and hemorrhagic foci; in a few cases, melanin can be recognized also melanin can impart dark-brown coloring or black discoloration [7,8]. On low-power examination, CCS presents a characteristic lobular growth pattern with fascicular/nested architecture, composed of plump fusiform to epithelioid cells partitioned by thin fibrous septa [11,12]. Typically, CCS has an infiltrative growth pattern and is made up of tumor cells with clear/eosinophilic cytoplasm, vesicular nuclei, and prominent basophilic nucleoli [13,14]. Although substantial nuclear pleomorphism and mitotic figures are absent, scattered wreath-like multinucleated giant cells are relatively common [4,14] and in a subset of cases there is the possibility to appreciate a junctional nested tumor component that can mislead the pathologist for differential diagnosis of malignant melanoma. Mitotic count of CCS is frequently low, usually <3 mitoses/mm^2^, and in 1/3 of cases is present necrosis [6,15]. Melanin is sometimes identified but often not abundant enough to be seen on H&E, and it can be detected with appropriate histochemical stains such as Fontana–Masson or Warthin–Starry and by electron microscopy which reveals the presence of melanosomes [15]. From an immunohistochemical point of view, S-100 protein, HMB-45, Melan-A, SOX-10, and MiTF are usually positive (melanocytic line differentiation features) [14,16,17] and recently also it was demonstrated the potential utility of negativity for PRAME in the differential diagnosis with melanoma [18]. The genetic hallmark of CCS is the reciprocal translocation t(12;22)(q13;q12) present in 70–90% of cases with the resulting fusion of EWSR1 with ATF. This event directly up-regulates microphthalmia transcription factor (MiTF) pathway with downstream activation of the pathway involved in melanocytic differentiation [3,4,5,6,7,8,9,10,11,12] and, notably, this translocation has not been documented in melanoma [19].

Figure 1A–D shows different histopathological fields of an example of CCS.

## 4. Focus on the Challenging Differential of Clear Cell Sarcoma

The differential diagnosis of CCS is a complex task due to the considerable overlap in histological and immunohistochemical features with other neoplasms, most notably melanoma [5,8]. Both CCS and melanoma share an expression of melanocytic markers, including S100, HMB-45, and Melan-A, as well as similar morphological characteristics, such as large epithelioid cells with clear cytoplasm and prominent nucleoli [19]. However, CCS displays distinctive features, such as a nodular growth pattern with spindle and epithelioid cells arranged in nests or sheets and the translocation t(12;22)(q13;q12), which results in the EWSR1-ATF1 fusion gene [20] that is the most reliable element for differential diagnosis. It is typically confirmed through Fluorescence In Situ Hybridization (FISH) or Real-Time Polymerase Chain Reaction (RT-PCR) testing [21]. Beyond melanoma, the differential diagnosis of CCS includes a variety of other neoplasms, particularly those with clear cell features. Renal cell carcinoma (RCC), for instance, may rarely metastasize to soft tissue sites and mimic CCS due to its clear cell morphology [22]. Immunohistochemically, RCC differs from CCS in that it is typically positive for markers such as CD10, PAX8, and EMA, while CCS lacks these markers [23]. Another diagnostic consideration is Alveolar Soft Part Sarcoma (ASPS), which, like CCS, tends to affect younger patients and may present with clear cell morphology. However, ASPS typically demonstrates a unique PAS-positive, diastase-resistant crystalline structure in its cytoplasm and shows a characteristic ASPSCR1-TFE3 fusion gene [24]. The emergence of PRAME expression as a potential biomarker adds a valuable tool for differentiation, as PRAME is more frequently and intensely expressed in melanoma than in CCS [25]. Indeed, recent studies indicate that high PRAME expression correlates strongly with malignant melanoma and can be useful in distinguishing melanoma from other clear cell tumors, including CCS [8]. When used in conjunction with traditional histological assessment and molecular studies, PRAME may enhance diagnostic accuracy and support a more tailored therapeutic approach. Employing a multimodal diagnostic approach, which integrates immunohistochemistry, molecular testing and PRAME evaluation, is essential to avoid misdiagnosis and ensure the most effective treatment pathway for patients [26]. In any case, it is important to underline that not all melanomas can be positive for PRAME and this cannot be the only feature on which to make the diagnosis.

Table 2 summarizes the key differences between CCS and mimickers.

### Novel Entities Similar to CCS Included in WHO Classification of Skin Tumors, Fifth Edition

In the latest WHO *Classification of Skin Tumors*, fifth edition, three new entities were described and added in two different chapters (melanocytic and soft tissue) of the classification and can be recognized as MITF pathway-activated cutaneous melanocytic tumors, and are represented by CRTC1::TRIM11 cutaneous tumors (malignant soft tissue tumors with uncertain differentiation), ACTIN::MITF cutaneous tumors (melanocytic), and MITF::CREM cutaneous tumors (melanocytic) [27,28].

Although they are quite different from CCS in terms of aggressiveness and clinical course, these three entities are quite similar in terms of histological and immunohistochemical characteristics. In particular, CRTC1::TRIM11 cutaneous tumors are included in malignant soft tissue tumors with uncertain differentiation and approximately 40 cases are reported in the literature so far, with the most frequent localizations being distal extremities and trunk. From a histological point of view, these are well-defined dermal papules or nodules, rarely in submucosal regions, with occasional involvement of subcutis, constituted by short intersecting fascicles/nests of relatively uniform epithelioid/spindle cells, eosinophilic cytoplasm, moderate/severe atypia, prominent nucleoli, and limited mitotic activity (<5 mitosis/10 HPF); usually there are no atypical mitosis and occasionally there are binucleated/multinucleated cells. Finally, necrosis areas are rare. Immunohistochemically, SOX10 is almost always positive, while 80% of cases are positive for S-100 protein (often limited in extent). At least 60% of cases are positive for HMB-45 and Melan-A and at least one of these markers is positive in 80% of cases. Immunostaining for MiTF is diffuse positive in majority of cases while pan-TRK diffuse cytoplasmic positivity is present in 57% of cases and partial cytoplasmic positivity in 36% despite lack of TRK mutation. TRIM11 is diffuse nuclear positive in 94% of cases and there is negativity for PRAME, cytokeratins, and SMA [27,28]. Molecular confirmation is mandatory for this kind of neoplasm and the main differential diagnoses are with cellular blue naevus, melanoma, CCS, PEComa, and also MITF::CREM tumor. Prognosis is relatively favorable, and out of 38 cases with follow-up, six cases showed local recurrence (3/6), lymph node metastasis (5/6), or pulmonary metastasis (3/6) [27].

## 5. Treatment

Surgery remains the primary treatment option for CCS, with the integration of sentinel lymph node biopsy suggested to enhance disease staging. Locoregional approaches, including radiation therapy and isolated limb perfusion, may help preserve the affected limb and prevent amputation. However, chemotherapy has shown limited efficacy in treating metastatic CCS. In the realm of targeted therapy, efforts to leverage EWSR1 gene rearrangement, which leads to MET overexpression, have thus far produced disappointing results. Meanwhile, the clinical application of immunotherapy is still in its early stages, requiring further investigation to determine its potential effectiveness [29] (Table 3).

### 5.1. Treatment of Localized Disease

#### 5.1.1. Surgery

Wide surgical excision remains the only curative option for localized CCS. Unfortunately, many patients initially undergo incomplete surgery, resulting in positive resection margins. The primary objective of surgery is to achieve a macro- and microscopically complete resection with negative margins, though this often necessitates aggressive and sometimes mutilating procedures. It is well established that tumor excision with R1 and R2 margins is linked to a significantly worse prognosis [17,30]. However, more aggressive surgical approaches do not necessarily reduce the risk of local recurrence or distant metastasis [31] and should only be considered when limb-sparing surgery is not a viable option. Locoregional treatments, such as radiotherapy [6] and isolated limb perfusion [31], can be employed as alternatives to prevent amputation.

#### 5.1.2. Lymphadenectomy and Sentinel Biopsy

Unlike other sarcomas, CCS metastasizes to regional lymph nodes in approximately one-third of patients [14,32,33]. The majority of patients with lymph node metastases eventually develop distant metastases, which significantly impact overall survival [15,34]. To facilitate early detection of lymph node involvement, the sentinel lymph node (SN) procedure is recommended for CCS. Originally established for melanoma patients [35,36], this technique may also be beneficial in CCS management. The key advantage of SN biopsy is that it targets only the earliest draining lymph nodes, which have the highest risk of harboring metastases. Pathologists initially analyze these nodes using routine H&E staining, and if no metastases are detected, additional immunohistochemistry is performed to identify micro-metastases. Early detection of micro-metastases enables timely completion of lymph node dissection, potentially improving local disease control and offering a survival benefit.

#### 5.1.3. Adjuvant Treatment

It is not yet clear whether adjuvant radiotherapy adds a survival benefit after surgical resection. The European Society of Medical Oncology (ESMO) guidelines recommend adjuvant radiotherapy in cases of high-grade soft tissue sarcomas (grade 2–3) and large (>5 cm) deep soft tissue sarcomas (level IIB) [37]. The classification of soft tissue sarcomas is based on the Fédération Nationale des Centres de Lutte Contre le Cancer (FNCLCC), which distinguishes three grades of malignancy based on differentiation, necrosis, and mitotic rate. However, CCS typically has low mitotic activity and necrosis is only occasionally seen in these tumors [38], classifying most CCS as grade 1 tumors. However, adjunctive radiotherapy should be considered whenever resection margins are negative and additional surgery is not feasible or is refused by the patient. In general, the data on the adjuvant use of chemotherapy in STS are controversial, and it cannot be recommended in CCS, which is known to be a histological subtype of sarcoma that is very resistant to chemotherapy.

### 5.2. Treatment of Metastatic Disease

#### 5.2.1. Chemotherapy

Despite optimal management of localized disease, a high percentage of patients develop metastatic disease [15,39]. Although chemotherapy is mainly used in patients with metastatic disease, the treatment of advanced CCS remains challenging due to the lack of an established standard treatment system. Anthracyclines are widely administered in metastatic CCS, with few reports of partial responses (4%) [40]. However, other reports have not demonstrated the efficacy of anthracycline-based chemotherapy [33,39]. Another study reported partial responses in one-third (1/3) of patients who received chemotherapy, albeit with different regimens, suggesting increased chemosensitivity to CCS [29]. Finally, one author described a case of metastatic CCS with complete remission following combination chemotherapy with dacarbazine, nimustine, and vincristine, a treatment scheme used for malignant melanoma [13]. These previous studies suggest that alternative strategies are needed for systemic therapy of CCS. Although there is no consensus on the roles of radiotherapy and chemotherapy, these are unlikely to be promising treatments for CCS. In this context, the development of new therapeutic approaches, such as ICI and molecule-targeted therapies, is a step toward improving the survival of CCS patients.

#### 5.2.2. Target Therapy

As mentioned above, attempts to exploit the rearrangement of the EWSR1 gene (which leads to MET overexpression) are the basis of target therapy for CCS. The identification of features of EWSR1-ATF1 genesis has led to the development of targeted therapies, such as MET inhibitors. The efficacy of a tyrosine kinase inhibitor (TKI) was evaluated in a phase 2 study for MET-positive, locally advanced, or metastatic CCS [42]. The study demonstrated that the TKI provides some clinical benefit; however, it did not achieve the primary endpoint (global response), as partial response was observed only rarely [43].

#### 5.2.3. Immunotherapy

There is growing interest in investigating the potential of immune checkpoint inhibition in the treatment of CCS; pembrolizumab has been reported to be effective in a case of relapsed CCS [13]. Like malignant melanoma, CCS is associated with the expression of MITF, which is involved in tumor growth. Because these diseases have been hypothesized to share similarities in host immune reactivity toward tumors, nivolumab is expected to be a promising treatment for CCS as already successfully described in melanoma [44,45,46].

#### 5.2.4. Limitations of Current Therapeutic Approaches

CCS remains a challenging malignancy with limited therapeutic options. Surgical resection is the primary treatment for localized disease; however, high recurrence rates and distant metastases significantly compromise long-term survival. Although adjuvant radiotherapy is often used for cases with positive surgical margins, its impact on overall survival remains unclear. Chemotherapy, primarily based on anthracycline and ifosfamide regimens, has shown disappointing results, with objective response rates below 10% in metastatic CCS patients. Given the lack of efficacy of conventional treatments, alternative approaches are urgently needed to improve patient outcomes.

#### 5.2.5. Limitations

Being a narrative review, this work presents limitations due to its intrinsic methodology; therefore, systematic reviews of the literature are urgent to improve knowledge of this rare entity as well as to increase the search for more effective therapeutic treatments.

## 6. Conclusions

CCS remains a rare yet highly malignant neoplasm that presents substantial diagnostic and therapeutic challenges. Its resemblance to melanoma, both morphologically and immunohistochemically, underscores the critical need for accurate differentiation through molecular diagnostics, particularly the identification of the EWSR1-ATF1 fusion gene. The incorporation of novel immunohistochemical markers, such as PRAME, offers a promising avenue to reduce diagnostic ambiguity and improve precision in distinguishing CCS from melanoma and other histologically similar neoplasms.

The clinical presentation and prognosis of CCS highlight its aggressive nature, with high rates of metastasis and recurrence despite optimal management of localized disease. Wide surgical excision with negative margins remains the cornerstone of treatment for localized CCS, while the role of adjuvant therapies, including radiotherapy and chemotherapy, remains limited and controversial. For metastatic disease, traditional chemotherapy has demonstrated minimal efficacy, emphasizing the need for innovative approaches.

Emerging therapies, including targeted treatments focusing on MET inhibitors and immune checkpoint inhibitors, offer new hope but require further validation through clinical studies. The exploration of immunotherapy, such as pembrolizumab and nivolumab, draws parallels with advancements in melanoma treatment and represents a potentially transformative approach for CCS.

Overall, a multimodal strategy integrating advanced diagnostic tools, precise surgical techniques, and novel systemic therapies holds promise for improving outcomes in CCS patients. Continued research and collaborative efforts are essential to address the unmet needs in the diagnosis and treatment of this challenging malignancy.

## Figures and Tables

**Figure 1 jcm-14-01233-f001:**
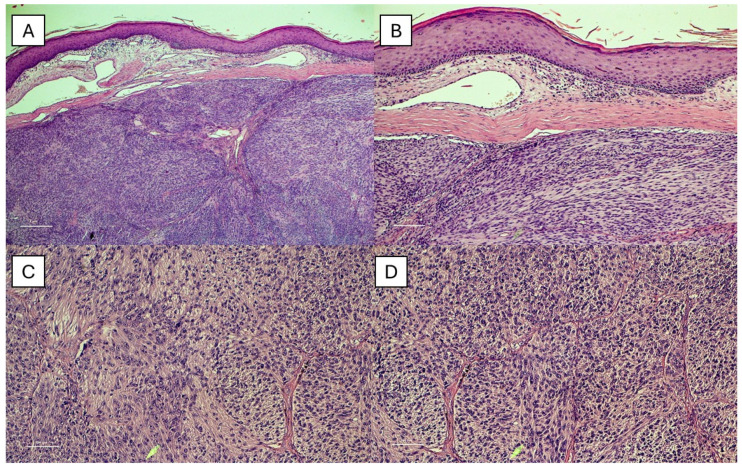
(**A**) Histopathological photomicrograph showing a solid dermal nodule (extending from deep soft tissue) without junctional component (H&E, original magnification 4×). (**B**) Higher magnification showing no contact between tumor and overlying epidermis (H&E, original magnification 10×). (**C**,**D**) Histological photomicrographs showing nests and strands of relatively bland oval and elongated cells separated by thin collagenous septa; there are large amounts of pale/granular amphophilic cytoplasm. Mitotic figures are rare (H&E, original magnification 20×). (Original pictures.).

**Table 1 jcm-14-01233-t001:** Table 1 summarizes some clinical-pathological features.

Age (Year)	Sex	Location	Survival Rates
20–40, median 30	Slight female predominance	40% ankle/foot 30% knee, thigh, and hand	5-year survival: 52% 10-year survival: 25%

**Table 2 jcm-14-01233-t002:** Summary of the most important key differences between CCS, melanoma, RCC, and ASPS.

Key Differences	Morphology	IHC	Molecular Biology
Clear Cell Sarcoma	Nodular growth pattern with spindle and epithelioid cells arranged in nests or sheets	S100, HMB-45, and Melan-A positivity	t(12;22)(q13;q12), resulting in EWSR1-ATF1 fusion gene
Melanoma	Large epithelioid cells with clear cytoplasm and prominent nucleoli similar to CCS	S100, HMB-45, and Melan-A positivity	BRAF, etc.
Renal Cell Carcinoma	Clear cell morphology	CD10, PAX8, and EMA positivity	Usually not mandatory
Alveolar Soft Part Sarcoma	PAS-positive, diastase-resistant crystalline structure in the cytoplasm	TFE3 and Cathepsin K	ASPSCR1-TFE3 fusion gene

Legend. IHC: immunohistochemistry.

**Table 3 jcm-14-01233-t003:** Treatment overview.

Treatment	Purpose	Effects	Limitations	Ref.
Surgery	Curative for localized CCS; requires complete resection.	Effective if negative margins are achieved; reduces recurrence.	High recurrence risk; may require aggressive/mutilating surgery.	[26,27]
Sentinel LN Biopsy	Detects early lymph node metastases; aids staging.	Helps determine disease spread and treatment planning.	Not always needed; role in CCS is still under investigation.	[6,14,17,30,31,32,33]
Radiotherapy	Adjuvant therapy to improve local control in high-risk cases.	May prevent local recurrence in select high-risk patients.	Uncertain survival benefit; limited impact in low-grade CCS.	[15,34]
Chemotherapy	Systemic treatment for metastatic CCS.	May induce partial response in some cases.	Poor response rates (<10%); common resistance.	[31,35,36,37,38]
Targeted Therapy	Targets MET overexpression; experimental approach.	Limited clinical benefit observed in trials.	Partial responses; no established long-term benefit.	[39]
Immunotherapy	Checkpoint inhibitors may enhance immune response in CCS.	Potential efficacy in CCS, similar to melanoma therapy.	Low PD-L1 expression may limit effectiveness.	[13,40,41]

## Data Availability

Data are contained in the manuscript.

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
