# Peer review of "Clear Cell Sarcoma (CCS) of the Soft Tissue: An Update Narrative Review with Emphasis on the Utility of PRAME in Differential Diagnosis"

_jcm, 2025, doi:10.3390/jcm14041233_

Round 1
Reviewer 1 Report
Comments and Suggestions for Authors
Manuscript entitled "Clear Cell Sarcoma (CCS) of the Soft Tissue: An Update Narrative Review with Emphasis on Utility of PRAME in the Differential Diagnosis" by Gerardo Cazzato et al.
This manuscript provides an updated review of Clear Cell Sarcoma (CCS), with a particular focus on the diagnostic utility of PRAME. It synthesizes current knowledge on CCS’s clinical presentation, histopathology, and emerging therapies. While the topic is relevant and the manuscript is well-structured, several areas need further improvement to enhance clarity and impact.
Comments:
1. In introduction section provide a more detailed explanation of PRAME’s functional role in melanocytic tumors and its relevance in differentiating CCS from melanoma. Further, highlight how integrating PRAME and molecular diagnostics can reduce diagnostic delays and improve patient outcomes.
2. The literature search strategy is briefly described but could be strengthened by justifying the inclusion of certain studies and the exclusion of others.
3. The section on clinical features and prognosis should include a table summarizing key clinical characteristics, such as age, sex, location, and recurrence rates, and expand the discussion to explore potential prognostic markers beyond tumor size and stage.
4. The section on histopathology and differential diagnosis should include a comparison table that outlines the key differences between CCS, melanoma, and other mimicking neoplasms.
5. The section on therapeutic strategies should elaborate on the limitations of existing therapies and provide a more detailed discussion of targeted therapy and immunotherapy. Additionally, a dedicated part should focus on emerging studies that analyze new therapeutic modalities.
6. The section on literature search methodology should include a systematic presentation of the findings from the literature. It should provide details on the total number of articles initially identified, the number included after screening for relevance, and the types of studies reviewed, such as case reports, retrospective analyses, and clinical trials. Furthermore, the addition of a table summarizing key studies would enhance clarity. This table should include information such as the author, year, and country; study design (e.g., case report, cohort study); patient population; diagnostic markers analyzed (e.g., PRAME, EWSR1-ATF1 fusion); and key findings.
7. The manuscript does not critically address gaps in the current literature.
8. Figure 1 requires the inclusion of a scale bar. Additionally, the figure lacks details regarding its origin, making it unclear whether it was created for this study or adapted from another source.
9. The review should explicitly address its limitations.
Author Response
Reviewer n’1: In introduction section provide a more detailed explanation of PRAME’s functional role in melanocytic tumors and its relevance in differentiating CCS from melanoma. Further, highlight how integrating PRAME and molecular diagnostics can reduce diagnostic delays and improve patient outcomes.
Answer n’1: Done dear Reviewer n’1. We have added the following sentences: “More in detail, PRAME firstly was isolated by autologous T-cells clone of a patient with metastatic cutaneous melanoma and in the last years was employed as marker to help dermatopathologists with the differential diagnosis between high grade dysplastic nevus and melanoma in situ, beyond that with other types of melanocytic lesions [7-8]. Furthermore, a recent study showed that PRAME was significantly up-regulated in a large number of tumors, and its high expression was correlated with poor overall survival across many cancer types, based on a pan-cancer survival analysis. Additionally, in a number of malignancies, PRAME expression levels were closely associated with immunological neoantigens, immune infiltration, immune score, immune checkpoint, tumor mutation burden, microsatellite instability, mismatch repair, and DNA methyltransferase. Finally, PRAME was linked to the control of many signaling pathways implicated in tumor immunity and tumorigenicity, according to GSEA analysis [9]”.
Reviewer n’1: The literature search strategy is briefly described but could be strengthened by justifying the inclusion of certain studies and the exclusion of others.
Answer n’2: Thank you. We have added some further explanations.
Reviewer n’1: The section on clinical features and prognosis should include a table summarizing key clinical characteristics, such as age, sex, location, and recurrence rates, and expand the discussion to explore potential prognostic markers beyond tumor size and stage.
Answer n’3: We have added a summarizing table (table 1) and also these sentences “Furthermore, a recent study addressed the issue of prognostic markers of CCS, analyzing data from the Surveillance, Epidemiology, and End Results (SEER) database, representing approximately 30% of the US population, over the time period 1973-2009. In particular, the authors extrapolated the demographic and clinical-pathological data of 175 patients diagnosed with CCS, evaluating a 5-year survival rate of 62.9% and 10-year survival rate of 51.3%. According to these data, 98 patients (56.0%) were men and 77 patients (44.0%) were women, with 101 patients (57.71%) with localized CCS, 59 patients (33.71%) with regional CCS and 15 patients (8.57%) with distant/metastatic CCS. Most common location at diagnosis was lower extremities (74.29%) and the tumor size most commonly was </= 3 cm at the time of diagnosis (44.57%) with 174 patients (93.71%) underwent surgical resection. Performing univariate Cox regression analysis, factors as race, sex, tumor stage, tumor size, chemotherapy, and surgery were statistically significant, but with the multivariate Cox regression analysis only size and tumor stage were significant, with a higher risk of death among patients with distant stage and regional stage and patients with tumor size >3 cm [10]”.
Reviewer n’1: The section on histopathology and differential diagnosis should include a comparison table that outlines the key differences between CCS, melanoma, and other mimicking neoplasms.
Answer n’4: Done.
Reviewer n’1: The section on therapeutic strategies should elaborate on the limitations of existing therapies and provide a more detailed discussion of targeted therapy and immunotherapy. Additionally, a dedicated part should focus on emerging studies that analyze new therapeutic modalities.
Answer n’5: Done, and we have also added a summarizing table (Table 3).
Reviewer n’1: The section on literature search methodology should include a systematic presentation of the findings from the literature. It should provide details on the total number of articles initially identified, the number included after screening for relevance, and the types of studies reviewed, such as case reports, retrospective analyses, and clinical trials. Furthermore, the addition of a table summarizing key studies would enhance clarity. This table should include information such as the author, year, and country; study design (e.g., case report, cohort study); patient population; diagnostic markers analyzed (e.g., PRAME, EWSR1-ATF1 fusion); and key findings.
Answer n’6: Thank you dear Reviewer n’1. We have added some features to “material and methods” section, and we think that in a narrative review could be sufficient. Obviously, if you disagree with us we can improve also this aspect.
Reviewer n’1: The manuscript does not critically address gaps in the current literature.
Answer n’7: Dear Reviewer n’1, we tried to improve this aspect. Thank you.
Reviewer n’1: Figure 1 requires the inclusion of a scale bar. Additionally, the figure lacks details regarding its origin, making it unclear whether it was created for this study or adapted from another source.
Answer n’8: Done dear Reviewer. Figure 1 is totally original. We have added this information in the manuscript.
Reviewer n’1: The review should explicitly address its limitations.
Answer n’9: Done. Thank you.
Reviewer 2 Report
Comments and Suggestions for Authors
abstract
- line 32: how sarcoma resembles melanoma
intro
- what about clinical and dermoscopy aspects of cc sacrcomas
methods
-please provide the images in the correct order
- also provide a table with histopathologic findings of your samples
in the discussion section- you report about treatments on ccsarcomas
what is the role of PRAME in choosing treatment??
can there be PRAME-false posirive and negarive samples? yes or no and why-
Author Response
Reviewer n’2: line 32: how sarcoma resembles melanoma.
Answer n’1: Thank you. From a morphological and immunohistochemical point of view.
Reviewer n’2: intro
- what about clinical and dermoscopy aspects of cc sacrcomas
Answer n’2: Thank you. We have added the following sentence: “with a clinical appearance of a palpable mass, sometimes with pain and tenderness”.
Reviewer n’2: methods
-please provide the images in the correct order
- also provide a table with histopathologic findings of your samples
Answer n’3: We have corrected the order of the pictures and we have added also a table (table 2) with the most important clinical and pathological features of the CCS.
Reviewer n’2: in the discussion section- you report about treatments on ccsarcomas
what is the role of PRAME in choosing treatment??
Answer n’4: Thank you very much for the question. At the moment there are no available informations about the use of PRAME for the choice of the treatment.
Reviewer n’2: can there be PRAME-false posirive and negarive samples? yes or no and why-
Answer n’5: Thank you for this question. We have added this sentence: “Anyway, it’s important to underline that not all melanoma can be positive for PRAME and this cannot be the only features on which to make the diagnosis”.
Round 2
Reviewer 1 Report
Comments and Suggestions for Authors
The authors have adequately addressed my comments, and the manuscript can be accepted for publication.
Reviewer 2 Report
Comments and Suggestions for Authors
the authors responded adequately to my concerns and the manuscripr was improved